# The Effects of Swine Coronaviruses on ER Stress, Autophagy, Apoptosis, and Alterations in Cell Morphology

**DOI:** 10.3390/pathogens11080940

**Published:** 2022-08-19

**Authors:** Ya-Mei Chen, Eric Burrough

**Affiliations:** 1College of Veterinary Medicine, National Pingtung University of Science and Technology, Neipu, Pingtung County 912301, Taiwan; 2Department of Veterinary Diagnostic and Production Animal Medicine, College of Veterinary Medicine, Iowa State University, Ames, IA 50011, USA

**Keywords:** coronaviruses, ER stress, UPR, autophagy, apoptosis, epithelial-mesenchymal transition, tight junction proteins, cytoskeletal rearrangement

## Abstract

Swine coronaviruses include the following six members, namely porcine epidemic diarrhea virus (PEDV), transmissible gastroenteritis virus (TGEV), porcine delta coronavirus (PDCoV), swine acute diarrhea syndrome coronavirus (SADS-CoV), porcine hemagglutinating encephalomyelitis virus (PHEV), and porcine respiratory coronavirus (PRCV). Clinically, PEDV, TGEV, PDCoV, and SADS-CoV cause enteritis, whereas PHEV induces encephalomyelitis, and PRCV causes respiratory disease. Years of studies reveal that swine coronaviruses replicate in the cellular cytoplasm exerting a wide variety of effects on cells. Some of these effects are particularly pertinent to cell pathology, including endoplasmic reticulum (ER) stress, unfolded protein response (UPR), autophagy, and apoptosis. In addition, swine coronaviruses are able to induce cellular changes, such as cytoskeletal rearrangement, alterations of junctional complexes, and epithelial-mesenchymal transition (EMT), that render enterocytes unable to absorb nutrients normally, resulting in the loss of water, ions, and protein into the intestinal lumen. This review aims to describe the cellular changes in swine coronavirus-infected cells and to aid in understanding the pathogenesis of swine coronavirus infections. This review also explores how the virus exerted subcellular and molecular changes culminating in the clinical and pathological findings observed in the field.

## 1. Swine Coronaviruses

The family *Coronaviridae* consists of two subfamilies, *Letovirinae* and *Orthocoronavirinae* (Virus Taxonomy: 2020 Release (MSL #36); https://talk.ictvonline.org/taxonomy/ (accessed on 18 January 2022)). *Orthocoronavirinae* are comprised of four genera: *Alphacoronavirus*, *Betacoronavirus*, *Gammacoronavirus*, and *Deltacoronavirus*. Six swine coronaviruses (CoVs) have been identified: porcine epidemic diarrhea virus (PEDV), transmissible gastroenteritis virus (TGEV), porcine delta coronavirus (PDCoV), swine acute diarrhea syndrome coronavirus (SADS-CoV), porcine hemagglutinating encephalomyelitis virus (PHEV), and porcine respiratory coronavirus (PRCV) [1]. PEDV, TGEV, SADS-CoV, and PRCV belong to the *Alphacoronavirus* genus; PHEV belongs to the *Betacoronavirus* genus; and PDCoV belongs to the *Deltacoronavirus* genus [2].

Swine CoVs are enveloped, single-stranded, positive-sense RNA viruses, and the viral genome consists of open reading frame (ORF) 1a (ORF1a), ORF1b, HE, S, ORF3, E, M, ORF6, N, and ORF7 [1] (Figure 1). Among these genes, the S, E, M, and N genes encode structural proteins, including spike (S), envelope (E), membrane (M), and nucleocapsid (N) proteins, respectively. HE gene, only in PHEV, encodes hemagglutinin esterase. ORF1ab encodes 15–16 nonstructural proteins, and ORF 3, ORF6, and ORF7 encode accessory proteins. PEDV, TGEV, PDCoV, and SADS-CoV are enteropathogenic, leading to vomiting, diarrhea, and dehydration in pigs of all ages, especially in neonatal piglets. These enteropathogenic coronaviruses replicate in absorptive epithelial cells of the intestine, mainly in the jejunum and ileum, which later results in apoptosis, necrosis, and sloughing of epithelial cells [3,4,5,6] (Figure 2). PHEV and PRCV cause nervous disorder and respiratory diseases, respectively (Table 1).

## 2. ER Stress

The endoplasmic reticulum (ER) in the cytoplasm of eukaryotic cells is responsible for protein synthesis and folding. ER stress is a condition in which ER homeostasis is disrupted, resulting in the accumulation of unfolded or misfolded proteins in the ER. Various physiological and pathologic factors can induce ER stress, such as gene mutations, hypoxia, nutrient deprivation, cell injury, and pathogen infection [9]. Cells initiate the unfolded protein response (UPR) to restore ER homeostasis [10]. Through a series of signal transduction pathways, the UPR removes aberrant proteins by inhibiting protein translation, increasing protein folding capacity, and promoting ER-associated degradation (ERAD) [11]. Three protein sensors, namely pancreatic ER eIF2α kinase (PERK), activating transcription factor 6 (ATF6), and inositol-requiring transmembrane kinase/endonuclease 1 (IRE1), play critical roles in the UPR [11].

In the homeostatic ER, PERK, ATF6, and IRE1 are connected to the ER membrane by the ER chaperone glucose-regulated protein 78 (GRP78, also known as BiP) [11]. When the ER is stressed, GRP78 dissociates from the intraluminal domains of sensors into the ER lumen and activates these three proteins. Among the three sensors, PERK is activated firstly. Activated PERK phosphorylates the α subunit of eukaryotic translation initiation factor 2 (eIF2α), followed by the translation of activating transcription factor 4 (ATF4). As a transcription factor, ATF4 regulates genes involved in protein folding and the oxidative stress response. Second, activated ATF6 moves from the ER into the nucleus and stimulates UPR genes, resulting in elevated expression of X-box binding protein 1 (XBP1) and GRP78. As a result, an increased amount of GRP78 is considered the hallmark of ER stress and UPR [12]. Later, activated IRE1 splices XBP1 mRNA to generate a functionally active isoform of XBP1 (XBP1s), which is a transcription factor that regulates most UPR-associated genes [13]. Additionally, ER stress is associated with autophagy (Figure 3), which contributes to removing unnecessary or dysfunctional cellular components. For example, IRE1 is required for autophagy activation [14]. Therefore, the UPR induces a pro-survival adaptation via the PERK, ATF6, and IRE1 pathways.

When ER stress is persistently unresolved, cells may undergo apoptosis or chronic ER stress (Figure 3). Apoptosis is a type of programmed cell death, and ER stress-induced apoptosis is responsible for eliminating cells under irremediable ER stress. For example, ATF4 upregulates C/EBP homologous protein (CHOP), which is a proapoptotic transcription factor [15]. The relationship between apoptosis and infection of swine coronaviruses is described in the section “Apoptosis in Swine CoV Infection” below. On the other hand, stressed but surviving cells can manage protein synthesis and adapt to chronic ER stress. For example, neoplastic cells under ER stress persistently express an elevated GRP78 level to adapt to a hostile microenvironment [16]. One of the cellular alterations associated with chronic ER stress is the epithelial-mesenchymal transition (EMT), which is described in the section “Cellular Morphologic Alterations in Swine CoV Infection” below.

## 3. ER Stress in Swine CoV Infection

As RNA viruses, CoVs replicate in the cytoplasm of host cells, and viral proteins are mainly assembled in the ER. CoVs form double-membrane vesicles (DMVs) and convoluted membranes (CMs), which directly arise from the ER and the Golgi complex [17]. CoVs also acquire lipid envelopes from host cells when the viral nucleocapsid buds from the endoplasmic reticulum–Golgi intermediate compartment (ERGIC). These processes induce the depletion of the ER membrane and disrupt ER morphology and function, leading to ER stress [18] (Table 2). The production of viral transmembrane glycoproteins, such as the 3a protein of severe acute respiratory syndrome coronavirus 1 (SARS-CoV-1), also leads to ER stress [19]. For now, SARS-CoVs are well known for their pandemic threat to humans. Studies have demonstrated that SARS-CoV-2 induces DMVs, zippered ER, ER stress, and UPR through all three UPR pathways [20,21,22]. Zippered ER is a rearrangement that forms a paired ER membrane with double-membrane spherules.

During viral replication, CoVs shut down cellular translation to benefit the synthesis of viral products. For example, viral N protein can interact with elongation factor 1α (EF1α), a major translation factor, leading to translational suppression [23]. It has also been reported that SARS-CoV-2 can modulate the expression of RNAs and proteins in host cells [24]. To assess the variances in virus-infected cells, scientists applied gene expression profiling and mass spectrometry, which can detect differentially expressed genes (DEGs) and differentially expressed proteins (DEPs), respectively. DEGs are a broad group, and common detectable DEGs include RNA, microRNA (miRNA), circular RNA (circRNA), and messenger RNA (mRNA).

In eight-week-old weaned pigs, PEDV induces ER stress and UPR in jejunal epithelial cells through all three UPR pathways [25] (Figure 4, Table 2). In Vero cells, PEDV facilitates ER membrane rearrangement, ER stress, and UPR via the PERK and ATF6 pathways [17,26]. Recently, it has been demonstrated that PEDV induces reactive oxygen species (ROS)-dependent ER stress via the PERK and IRE1 pathways in Vero cells [27]. Interestingly, the mutation in the PEDV E protein enhances ER stress and apoptosis in Vero cells [28]. In intestinal epithelial cells (IECs), the E and N proteins but not the M protein of PEDV lead to ER stress [29,30,31]. PEDV upregulates DEGs that transport proteins between ER and Golgi compartments in intestinal porcine epithelial cell line-J2 (IPEC-J2) cells [32,33]. In human embryonic kidney (HEK) 293T cells, PEDV ORF3 protein upregulates GRP78 and later activates the PERK-eIF2α signaling pathway and autophagy [34].

The infection of TGEV induces ER stress and UPR via all three UPR pathways [35]. The TGEV N protein alone induces ER stress in porcine intestinal epithelial cells, while nonstructural protein 7 (nsp7) does not increase GRP78 expression [36,37]. Interestingly, the UPR suppresses the replication of TGEV in swine testicular (ST) and IPEC-J2 cells through the PERK-eIF2α axis [35]. On the other hand, IRE1 facilitates TGEV replication by downregulating type I interferon (IFN), which is a critical antiviral signaling protein [38].

Similar to other CoVs, PDCoV upregulates the protein level of GRP78 and induces DMVs and zippered ER formation [39,40]. A recent study has demonstrated that PDCoV induces ER stress through all three UPR pathways [41]. Few studies focus on SADS-CoV. SADS-CoV enhances the stress-associated RNA expression in IPEC-J2 cells, implying the possibility of disrupted ER homeostasis [42].

Differ from enteric coronaviruses, PHEV replicates in the cell bodies of neurons, and viral particles bud from the ERGIC [43]. PHEV also induces ER stress and all three branches of the UPR pathway both in vivo and in vitro [44]. Interestingly, Rab3a, a Golgi-associated protein, can act as a regulator of PHEV replication, indicating the close relationship between ERGIC and viral replication [45]. There is no published research focused on ER stress in PRCV infection at this time.

In addition to reinstatement of ER homeostasis, the UPR can modulate viral replication and the host’s innate response. The UPR suppresses PHEV replication via the PERK/PKR-eIF2α pathways [44]. PERK signaling also suppresses the replication of PEDV and TGEV [26,35]. Unsurprisingly, viruses have evolved strategies to counteract the UPR. For example, TGEV protein 7 accelerates eIF2α dephosphorylation to counteract host defense [46].

Stressed cells form stress granules (SGs), which are nonmembranous cytosolic RNA granules generated within the cytoplasm and closely associated with phosphorylated eIF2α [47]. The fundamental components of SGs are Ras-GTPase-activating protein-binding protein 1 (G3BP1), T-cell intracellular antigen 1 (TIA1), and poly(A) binding protein (PABP). It has been demonstrated that PHEV induces transient SG formation in the late stages of infection [44]. Increasing evidence suggests that SGs have the antiviral ability to limit viral replication. For instance, G3BP1 impairs PEDV replication in Vero cells [48]. However, PEDV suppresses the formation of SGs by inducing caspase-8-mediated G3BP1 cleavage [49]. Caspases are involved in apoptosis and discussed in the “Apoptosis in Swine CoV Infection” section.

**Table 2 pathogens-11-00940-t002:** Molecular and subcellular pathology caused by swine coronaviruses.

Name of Virus	ER Stress	Autophagy	Apoptosis	Alterations in Cell Morphology
PEDV	*In vivo* Jejunal epithelial cells in 8-week-old pigs [25] *In vitro* Vero cells [17,26,27] IECs [29,30,31] HEK293T cells [34]	*In vitro* Vero cells [27,34,50] IPEC-J2 cells [51,52] ST cells [53] HEK293T cells [54]	*In vivo* Small intestine in 4-day-old piglets [55] Jejunal epithelial cells in 5- and 8-week-old pigs [25,56] *In vitro* Vero cells [55,57,58,59,60] IECs [61] IPEC-J2 cells [62,63]	*In vivo* Reduced ZO-1 in jejunum in 4-week-old pigs [64] EMT in jejunum in 4-week-old pigs [64] *In vitro* Reduced ZO-1 in IPEC-J2 cells [65] Disrupted protein level of AJs and TJs in Vero cells [59]
TGEV	*In vitro* Porcine intestinal epithelial cells [35] ST cells [35]	*In vitro* IPEC-J2 cells [66]	*In vitro* IPEC-J2 cells [67] PK-15 cells [68,69,70,71,72] ST cells [73,74,75]	*In vitro* Reduced E-cadherin, occluding, ZO-1 in IPEC-J2 cells [65] Microfilament and F-actin reorganization in IPEC-J2 cells [65] EMT in IPEC-J2 cells [76] Elevated microfilament and microtubule in ST cells [77] Altered cytoskeleton and vimentin in ST cells [78]
PDCoV	*In vitro* IPI-2I cells [41] LLC-PK1 cells [41] PK-15 cells [39]	*In vitro* LLC-PK1 cells [79]	*In vivo* Jejunal and ileum in 7-day-old pig [80] *In vitro* LLC-PK1 cells [81,82] ST cells [81,82,83]	*In vivo* Reduced ZO-1 in small intestine in 7-day-old pigs [80]
SADS-CoV	*In vitro* IPEC-J2 cells [42]	n/a	*In vitro* Vero cells [84] IPI-2I cells [84]	n/a
PHEV	*In vivo* Mouse brains [44] *In vitro* N2a cells [44]	*In vitro* N2a cells [85,86]	*In vitro* PK-15 cells [87]	*In vivo* Actin rearrangement in mouse brain [88] *In vitro* Actin rearrangement in N2a cells [89,90]
PRCV	n/a	n/a	n/a	n/a

Abbreviations: AJs, adherens junctions; EMT, epithelial-mesenchymal transition; HEK293T cells, human embryonic kidney 293T cells; IECs, intestinal epithelial cells; IPEC-J2 cells, intestinal porcine epithelial cell line-J2; IPI-2I cells, porcine intestinal epithelial cells; LLC-PK1 cells, porcine kidney epithelial cells; n/a, not applicable; N2a cells, mouse neuro-2a cells; PK-15 cells, porcine kidney cells; ST cells, swine testicular cells; TJs, tight junctions; ZO-1, zonula occludens-1.

## 4. Autophagy in Swine CoV Infection

Autophagy is a self-degradative and highly regulated process in response to balance energy sources. Autophagy is also essential for removing misfolded or aggregated proteins, clearing damaged organelles, and eliminating intracellular pathogens [91]. Initially, the phagophore, an isolated membrane, engulfs cytosolic components and sequesters the intracellular cargo into an autophagosome. The autophagosome, a double-membrane vesicle, later fuses with lysosomes, promoting proteolytic degradation of cytosolic components. The autophagy-related complex consists of ULK1 protein kinases, autophagy-related gene 5 (Atg5)–Atg12 conjugation system, light chain 3 (LC3) conjugation system, adaptor protein p62/SQSTM1, and ubiquitinated proteins. Beclin-1 (BECN1) at the ER and other membranes contribute to phagophore formation [91]. It has been shown that SARS-CoV-2 induces autophagy because of excessive oxidative stress and ER stress [20]. The nsp6 protein of infectious bronchitis virus (IBV), an avian coronavirus, can also trigger autophagy [92]. Moreover, the attenuation of autophagy reduces SARS-CoV-1 and coronavirus mouse hepatitis virus (MHV) infection [93,94].

Whether swine CoVs inhibit or “hijack” autophagy for in vivo viral replication remains unclear (Table 2, Figure 5). PEDV, TGEV, PDCoV, and PHEV may induce autophagy to benefit viral replication [50,66,79,85]. PEDV leads to autophagy in Vero cells, and silence of the BECN1 or ATG5 gene reduces the viral titer, suggesting a potential profit from autophagy [50]. Additionally, the PEDV nsp6 protein induces autophagy by inhibiting the PI3K/Akt/mTOR pathway, which promotes cell death and attenuates autophagy [51,52]. In Vero cells, PEDV ORF3 protein induces the conversion of LC3-I to LC3-II, followed by autophagy. Moreover, the process is impaired by an ER stress inhibitor, indicating that PEDV ORF3 protein-induced autophagy is dependent on the ER stress [34]. Notably, PEDV induces autophagy in Vero cells via both the PERK and IER1 pathways, illustrating the relationship between ER stress and autophagy [27]. In ST cells, PEDV regulates miRNA and mRNA associated with the PI3K-Akt signaling pathway, implying the manipulation of autophagy [53]. In HEK293T cells, knockdown of the BECN1 gene decreases the PEDV replication [54]. Interestingly, autophagy can promote immunological defense mechanisms to counteract viral infection [95]. For instance, BECN1 negatively regulates the immune response by preventing excessive production of IFN [96]. As a result, PEDV negatively regulates the IFN pathway by inducing autophagy [54].

In vivo study has demonstrated that TGEV facilitates mitophagy, a selective autophagy in mitochondria, to counteract oxidative stress and apoptosis, suggesting that autophagy benefits viral replication [66]. Interestingly, doxycycline can induce mitophagy and facilitates TGEV replication [97]. PDCoV leads to autophagosome-like vesicles and autophagy by activating both LC3 I/II and p62 [79]. PDCoV also regulates DEPs involved in the PI3K/AKT/mTOR signaling pathway in porcine intestinal epithelial cells [98]. PHEV promotes ULK1-independent autophagy in mouse neuro-2a (N2a) cells, which is regulated by BECN1 [85]. Interestingly, PHEV induces autophagy but blocks the fusion of phagosomes and lysosomes to benefit viral replication [86]. PHEV also facilitates transcription factor EB (TFEB) to promote autophagosome formation [99]. It has been suggested that lysosomal dysfunction and defects in fusion are involved in the pathogenesis of PHEV infection [100].

Nevertheless, there are controversial data regarding the role of autophagy in swine CoV infection. MYH1485, an autophagy inhibitor, enhances the PEDV infection rate in Vero E6 cells [58]. Rapamycin, an autophagy inducer, impairs the infectivity of PEDV, TGEV, and PHEV but increases the expression of PDCoV [86,101,102,103]. In addition, genetic inhibition by knockdown of LC3, ATG5, and ATG7 demonstrates that TGEV replication is negatively regulated by autophagy [102]. SADS-CoV upregulates DEGs associated with the autophagy pathway in IPEC-J2 cells but downregulates them in Vero E6 cells [42,104]. No study has focused on the relationship between PRCV and cellular autophagy.

## 5. Apoptosis in Swine CoV Infection

Apoptosis is a process of programmed cell death, and apoptotic cells are morphologically characterized by condensed nuclei, fragmented DNA, and cellular shrinkage. The essential regulator of apoptosis is caspases, which are a family of proteolytic enzymes, and myriad proapoptotic and antiapoptotic proteins regulate the activity of apoptosis [91]. For instance, p53 is a proapoptotic protein involved in the gene transcription of DNA repair and cell cycle arrest. In contrast to caspase-dependent apoptosis, caspase-independent apoptosis is induced by released mitochondrial proteins or lysosomal membrane permeabilization [105]. For example, apoptosis-inducing factor (AIF), a mitochondrial protein, induces caspase-independent apoptosis by translocating from the mitochondrial intermembrane space to the nucleus [106].

Apoptosis is initiated by the FasL- (extrinsic) and mitochondria- (intrinsic) mediated pathways [107]. The extrinsic pathway involves cell surface death receptors, such as the tumor necrosis factor (TNF) receptor gene superfamily or Fas (CD95). On the other hand, the intrinsic pathway is activated by nonreceptor-mediated stimuli that produce intracellular signals and permeabilize the mitochondrial outer membrane. As mentioned previously, prolonged ER stress triggers ER stress-dependent apoptosis via the p38 mitogen-activated protein kinase (MAPK) signaling pathway [108].

The infection of CoVs induces apoptosis through complex mechanisms, and apoptosis has different effects on viral replication. For instance, SARS-CoV-2 induces apoptosis via the extrinsic pathway [109]. Apoptosis can facilitate viral release and dissemination from infected cells. On the other hand, inhibition of apoptosis prevents premature cell death, allowing viral replication. CoVs can regulate the MAPK signaling pathway to facilitate viral replication. For example, PEDV and PDCoV manipulate the JNK1/2 and p38/MAPK pathways to benefit viral biosynthesis and regulate immune responses [110,111].

The infection of PEDV facilitates apoptosis (Table 2, Figure 6) through many different mechanisms. In vivo studies have demonstrated that PEDV induces apoptosis, enterocyte proliferation, and a reduced ratio of villus height to crypt depth in nursing and weaned piglets [25,55,56]. In Vero cells, PEDV induces apoptosis by the p53-PUMA signaling pathway but not by the p38 MAPK or SAPK/JNK signaling pathways, and apoptosis is evidenced by activated proapoptotic protein p53 and oxidative stress [57,112]. PEDV also elevates the expression of DEGs associated with apoptosis, the p53 signaling pathway, and the MAPK signaling pathway in Vero cells [57,58]. PEDV facilitates both caspase-independent and caspase-dependent apoptosis. In PEDV-infected Vero cells, AIF and AIF-associated DEPs activate caspase-independent apoptosis [55,57,59], while caspase-8 and -3 activate the caspase-dependent pathway [60]. Compared to the activation of caspase-3 by both extrinsic and intrinsic apoptotic pathways, caspase-8 is the initiator caspase of extrinsic apoptosis only. Interestingly, PEDV downregulates caspase-8 protein in early viral infection, suggesting a benefit for viral replication [59,113]. As mentioned in the “ER Stress in Swine CoV Infection” section, PEDV utilizes caspase-8 to cleave G3BP1,a component of stress granules, to promote viral replication in Vero cells [49]. These findings elucidate a complex relationship among PEDV, ER stress, and apoptosis. Caspase-6 and -7 also cleave PEDV N protein at the late stage of viral replication in Vero cells, but the role of cleavage in viral pathogenicity remains unclear [114]. In Vero cells, PEDV S protein is a critical inducer of apoptosis [115], while ORF3 protein inhibits apoptosis to enhance viral proliferation [116]. The ORF3 protein of PEDV has no effect on apoptosis in HEK293T cells [34]. PEDV also facilitates apoptosis in IECs [61], and apoptosis in IPEC-J2 cells is associated with the PI3K/AKT/mTOR pathway [62,63]. PEDV increases the expression of circRNAs and DEPs involved in apoptosis in IPEC-J2 cells [33,63]. In Vero cells and jejunum of piglets, PEDV induces the expression of interleukin-11 (IL-11), which is antiapoptotic and likely inhibits viral replication [117].

In brief, TGEV induces apoptosis via p53 and ROS, subsequently activating the p38 MAPK signaling pathway. As a result, this pathway activates both extrinsic and intrinsic pathways as well as both caspase-dependent and caspase-independent pathways [68,69,70,71,72,73,74]. TGEV damages mitochondria by upregulating cellular miRNA-4331, resulting in the opening of the mitochondrial permeability transition pore (mPTP) and subsequently to mitochondrial permeability [67,75]. TGEV also downregulates miRNA-27b, which is responsible for inhibiting apoptosis [118]. Similar to PEDV, the TGEV N protein is cleaved by caspase-6 and -7 during apoptosis [74]. It is controversial whether the N protein of TGEV located in mitochondria induces mitochondrial injury [66,119]. Interestingly, TGEV protein 7 inhibits apoptosis [46]. Host cells have developed strategies to eliminate the damage. For example, miRNA-222 in porcine kidney (PK-15) cells as well as mitophagy and circRNA circEZH2 in IPEC-J2 cells regulate the mitochondrial dysfunction during TGEV infection [66,67,120]. p53 inhibits TGEV replication via IFN signaling and promotes cell cycle arrest at the S and G2/M phases [72,121,122]. In vitro studies demonstrate that TGEV induces apoptosis in ST cells and PK-15 cells but not in intestinal epithelial cells [66,70,73,82,123]. As mentioned previously, TGEV induces autophagy to suppress oxidative stress and apoptosis but benefit viral infection in IPEC-J2 cells [66]. In addition, TGEV facilitates pyroptosis, which is a highly inflammatory form of programmed cell death in IPEC-J2 cells [124].

In vivo study shows that PDCoV induces apoptosis via the p38/MAPK signaling pathway in the small intestine of 7-day-old piglets [80]. In addition, PDCoV induces apoptosis in both LC porcine kidney cells and ST cells but not in IPEC-J2 cells [81,83,125,126]. PDCoV can activate DEGs associated with the apoptosis signaling pathway in human and pig intestinal epithelial cells [127]. SADS-CoV induces caspase-dependent extrinsic and intrinsic apoptosis [84]. At 24 h post inoculation (hpi), SADS-CoV-infected Vero E6 cells increase the expression of DEGs associated with negative regulation of the apoptotic process, implying that SADS-CoV inhibits apoptosis [104]. It has been demonstrated that PHEV leads to apoptosis via a caspase-dependent pathway [87]. There has been no study regarding cellular apoptosis in PRCV infection.

## 6. Cellular Morphologic Alterations in Swine CoV Infection

Structural and junctional proteins maintain cellular morphology and tissue structure. In the gastrointestinal tract, cell–cell adhesion among epithelial cells is determined by junctional proteins, including tight junctions (TJs), adherence junctions (AJs), and desmosomes. TJs are especially involved in paracellular permeability and contain occludin, claudins, zonula occludens (ZOs), tricellulin, cingulin, and junctional adhesion molecules (JAMs) [128]. The viral infection damages the intact intestinal epithelium. For instance, the human immunodeficiency virus reduces the expression of TJ proteins in intestinal epithelial cells, resulting in intestinal barrier dysfunction and chronic diarrhea [129]. Here, we discuss the major cellular morphologic changes in swine CoV-infected cells, including alterations in tight junctions and the cytoskeleton and EMT induced by chronic ER stress (Table 2).

The major component of cellular structural proteins is the cytoskeleton, and cytoskeletal filaments consist of actin filaments (also known as microfilaments), microtubules, and intermediate filaments. Actin and microtubules provide intracellular transport pathways to endogenous cargos, while intermediate filaments are responsible for mechanical stability. Viruses may alter the cellular morphology of infected cells during viral infection, leading to cytoskeletal rearrangement and even collapse [130]. For instance, SARS-CoV-2 modulates cytoskeleton components, including actin, microtubules, and intermediate filaments, in Calu-3 cells [131]. SARS-CoV-2 also regulates DEGs associated with cell junctions and the cytoskeleton in lung cells [132].

The infection of PEDV results in a lower expression of ZO-1 in enterocytes both in vivo and in vitro [64,65] (Table 2). In addition, the RNA expression of TJs, including occludin, claudin-1, claudin-4, claudin-5, ZO-1, and ZO-2, reduces in both PEDV-infected 7-day-old piglets and IPEC-J2 cells [133]. These findings suggest that PEDV disrupts the integrity of tight junctions. In the cytoplasm of infected cells, PEDV uses microtubule proteins, including dynein and kinesin-1, for viral transportation [134,135]. In Vero cells, PEDV disrupts the protein levels of the GTPase signaling pathway related to cytoskeletal changes and the protein levels of TJs and AJs [59]. In ST cells, PEDV regulates miRNAs and mRNAs associated with focal adhesion, endocytosis, and regulation of cytoskeletal activity [53].

In IPEC-J2 cells, TGEV reduces the protein levels of E-cadherin, occludin, and ZO-1, indicating that TGEV impairs the integrity of the epithelial barrier [65]. TGEV reorganizes microfilaments with an accompanying cell membrane rearrangement in IPEC-J2 cells [65]. TGEV binds to epithelial growth factor receptor (EGFR) and subsequently activates F-actin polymerization and reorganization, leading to F-actin gathering at the cell membrane [136]. In ST cells, which are mesenchymal in origin, TGEV upregulates the microfilament-associated proteins beta-actin and microtubule-associated alpha-tubulin and beta-tubulin to facilitate viral transport [77] but reduces the expression of actin-associated proteins, including filamin-A, filamin-B, microtubule-associated protein 4, and actin-related protein 2/3, suggesting cytoskeletal disruption [78]. In PDCoV infection, epithelial cells in the jejunum and ileum decrease the expression of ZO-1 with no effect on claudin-1 and occludin [80]. The N protein of PDCoV also enhances the protein expression of ezrin, which is a cytoskeletal protein [39].

It has been demonstrated that PHEV leads to rapid actin rearrangement, implying that the cytoskeleton is associated with viral endocytosis [89]. More specifically, PHEV induces cytoskeletal rearrangement by binding to integrin α5β1 and activating the α5β1-FAK signaling pathway in N2a cells and mice [88,90]. In addition, PHEV relies on microtubules and intermediate filaments of infected cells to propagate within and among nerve cells [137]. To date, no study has focused on the cellular morphology in SADS-CoV or PRCV infection.

A phenotype switching by which epithelial cells lose polarity but gain migratory and invasive properties is known as epithelial-mesenchymal transition (EMT). EMT is critical in embryo development, wound healing, pathological fibrosis, and tumor progression. During EMT, cells lose E-cadherin, a critical component in adherens junctions, but express mesenchymal markers, such as neural cadherin (N-cadherin) and vimentin. EMT is triggered by a series of transcription factors, including Snail, TWIST, and zinc-finger E-box-binding (ZEB) in intestinal epithelial cells [138]. Notably, proinflammatory cytokines, such as the transforming growth factor β (TGFβ) family and IL-17, facilitate Snail expression [139,140]. TGFβ is produced by stressed cells or inflammatory cells and contributes to tissue repair. In humans, EMT is highly involved in the pathogenesis of inflammatory bowel disease (IBD), which is characterized by diarrhea and persistent chronic inflammation, suggesting the role of EMT in the injured digestive tract [141]. TGFβ facilitates EMT in swine intestinal epithelial cells via the MAPK/ERK pathway [76]. Viral and bacterial infection leads to chronic inflammation and subsequent EMT in enterocytes. For instance, *Salmonella enterica* serovar Typhimurium facilitates EMT of intestinal cells to enhance its invasion [142]. SARS-CoV-2 induces EMT in alveolar epithelial cells and later fibrosis [143].

Chronic infection with PEDV leads to EMT in enterocytes of weaned piglets with increased protein levels of TGFβ, implying persistent injury caused by PEDV [64]. Additionally, PEDV upregulates DEGs associated with the TGFβ signaling pathway in IPEC-J2 cells [32,144]. Persistent TGEV infection induces EMT in enterocytes and later promotes the infection of *Escherichia coli* [76]. In addition, TGEV upregulates the production of TGFβ as a potential inducer of EMT [145]. TGEV regulates the expression of vimentin in ST cells [77], and knockdown of vimentin in ST cells impairs TGEV replication, indicating that vimentin is essential in viral replication [146]. It is unclear whether PDCoV induces EMT; however, the virus decreases both mRNA and protein expressions of cytoskeletal proteins, including beta-actin and α-actinin-4 (ACTN4), in PK cells [147]. Additionally, PDCoV affects DEGs associated with the TGFβ signaling pathway in both pig and human epithelial cells [127]. In contrast to enteropathogenic coronaviruses that induce structural damage in intestinal epithelial cells, PHEV mainly affects neurons that do not undergo EMT. To date, no study has focused on EMT in SADS-CoV or PRCV infection.

## 7. Conclusions

Swine coronaviruses induce enteric, respiratory, and nervous system diseases in pigs. This review reveals the cellular damage and alterations caused by swine CoVs. Functionally, EMT, alterations in cell morphology, and chronic inflammation (Table 2, Figure 3) prevent infected enterocytes from normally absorbing nutrients and therefore lead to the loss of water, ions, and proteins into the intestinal lumen. All other effects such as ER stress, autophagy, and apoptosis culminate in the degeneration and necrosis of infected cells. Thereby, these injuries provide a molecular foundation for the significant clinical and pathological changes (Table 1) and economic losses observed in swine production. More research is needed to understand swine CoVs given that coronaviruses have the capability for recombination and pandemic outbreaks. It is worth mentioning that limited research has focused on PRCV, which is likely underestimated due to the minimal degree of clinical disease. However, regarding the pandemic threat of SARS-CoV, it may be prudent to focus increased attention on PRCV.

## Figures and Tables

**Figure 1 pathogens-11-00940-f001:**
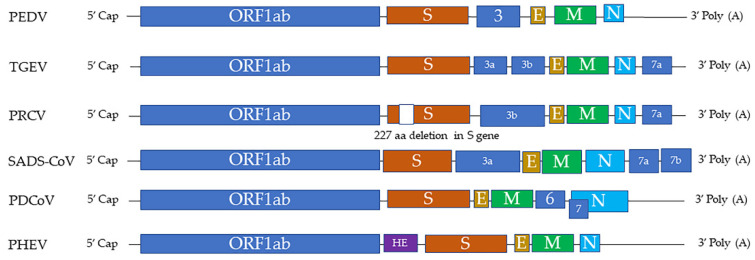
The genome structures of swine coronaviruses. ORF1ab, open reading frame genes 1a and 1b; S, spike; E, envelope; M, membrane; N, nucleocapsid; HE, hemagglutinin esterase; Ns3, Ns3a, Ns3b, Ns6, Ns7, Ns7a, Ns7b, accessory genes.

**Figure 2 pathogens-11-00940-f002:**
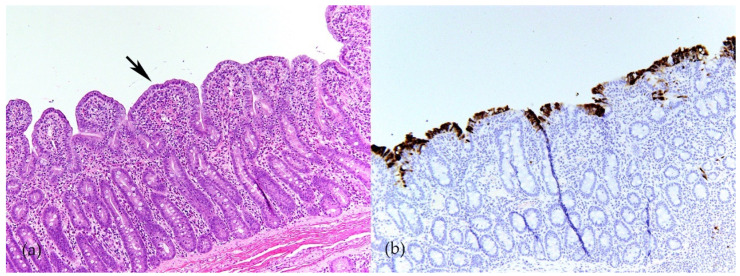
Porcine epidemic diarrhea virus (PEDV) infection in ileum of 4-week-old weaned pig. (**a**) Severe villous atrophy and villus fusion (arrow). Hematoxylin and eosin (HE). (**b**) Immunolabeling of PEDV in enterocytes (brown) indicates that PEDV mainly infect mature enterocytes on villi (Y-M Chen and E. Burrough, unpublished data, December 2017).

**Figure 3 pathogens-11-00940-f003:**
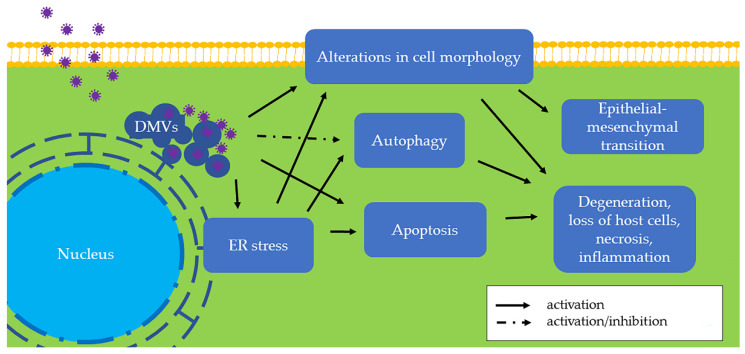
The effects of swine coronaviruses on host cells. Swine coronaviruses replicate in the cytoplasm of host cells and form double-membrane vesicles (DMVs), leading to endoplasmic reticulum (ER) stress, apoptosis, and alterations in cell morphology (Table 2). Swine coronaviruses may either induce or inhibit autophagy. These changes further evolve into degeneration, loss of host cells, necrosis, and inflammation. Furthermore, chronic alteration in cell morphology induces epithelial-mesenchymal transition (EMT). Some of these molecular and subcellular changes can be appreciated in vivo by pathological examination of tissues from infected swine, while others are found only in in vitro settings and remain to be found in clinical specimens (Table 1).

**Figure 4 pathogens-11-00940-f004:**
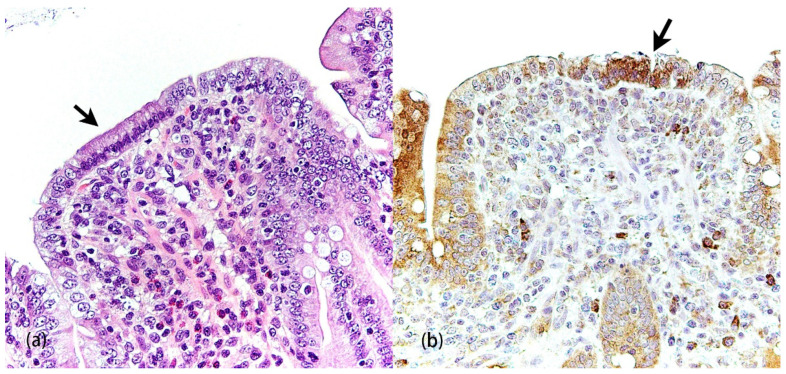
Porcine epidemic diarrhea virus (PEDV) infection in the jejunum of 8-week-old weaned pig. (**a**) Attenuated enterocytes (arrow) on an atrophic villus. Hematoxylin and eosin (HE). (**b**) Immunolabeling of GRP78 in attenuated enterocytes (arrow) indicates that PEDV infection leads to ER stress (Y-M Chen and E. Burrough, unpublished data, January 2021).

**Figure 5 pathogens-11-00940-f005:**
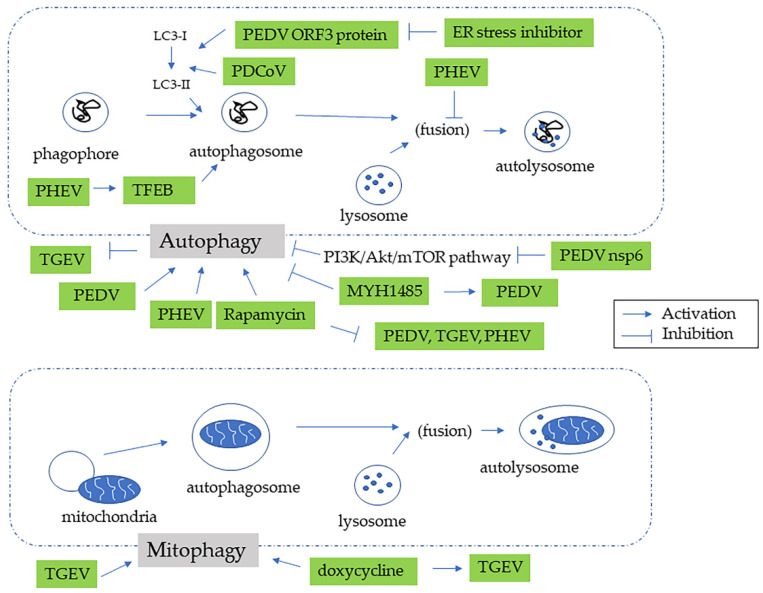
The relationship between autophagy and swine coronaviruses.

**Figure 6 pathogens-11-00940-f006:**
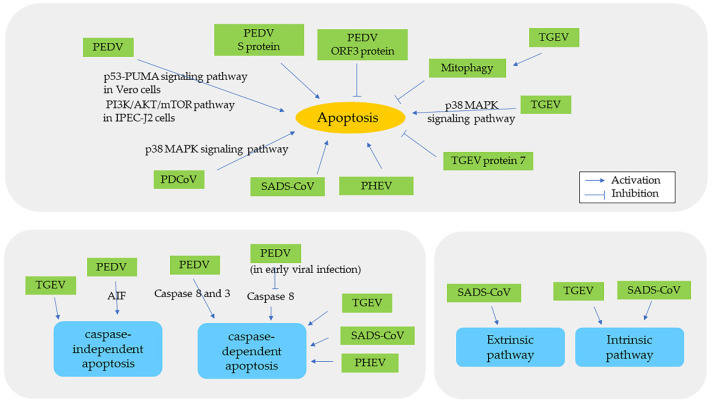
The relationship between apoptosis and swine coronaviruses.

**Table 1 pathogens-11-00940-t001:** Diseases caused by natural infections of swine coronaviruses.

Name of Virus/ Genera	Major System Affected	Summary and Noteworthy Clinical and Pathological Findings	Ref.
PEDV/ *Alphacoronavirus*	Enteric	Morbidity: 100% in piglets, less as pigs age Mortality: 50–100% in piglets ≤ 1 week of age Clinical signs: vomiting, watery diarrhea, anorexia, depression Gross lesions: distended small intestine containing yellow fluid and undigested milk Microscopic lesions: villous atrophy in the jejunum and ileum, necrosis of absorptive enterocytes in jejunum	[1]
TGEV/ *Alphacoronavirus*	Enteric	Morbidity: high morbidity in piglets ≤ 2 weeks of age, less as pigs age Mortality: up to 100% in piglets ≤ 2 weeks of age Clinical signs: similar to PEDV Gross lesions: similar to PEDV Microscopic lesions: similar to PEDV	[1]
PDCoV/ *Deltacoronavirus*	Enteric	Morbidity: up to 100% in piglets, less with age Mortality: up to 40% in suckling piglets Clinical signs: similar to PEDV Gross lesions: similar to PEDV and TGEV but less extensive Microscopic lesions: similar to PEDV and TGEV but less extensive	[1]
SADS-CoV/ *Alphacoronavirus*	Enteric	Morbidity: Up to 90% in piglets ≤ 5 days of age, less with age Mortality: over 35% in piglets ≤ 10 days of age Clinical signs: similar to PEDV Gross lesions: similar to PEDV and TGEV but less extensive Microscopic lesions: similar to PEDV and TGEV but less extensive	[7,8]
PHEV/ *Betacoronavirus*	Nervous	Morbidity: up to 100% in neonatal pigs Mortality: up to 100% in neonatal pigs Clinical signs: sneezing or coughing, nervous disorders, vomiting, wasting Gross lesions: cachexia, stomach dilatation, abdominal distension Microscopic lesions: nonsuppurative encephalomyelitis: perivascular cuffing, gliosis, and neuronal degeneration; most pronounced in the gray matter of the pons Varolii, medulla oblongata, and the dorsal horns of the upper spinal cord; degeneration of the ganglia of the stomach wall and perivascular cuffing	[1]
PRCV/ *Alphacoronavirus*	Respiratory	A variant of TGEV with a 227 aa deletion in S gene Morbidity: all ages of pigs can be infected Mortality: minimal (usually subclinical) Clinical signs: coughing, abdominal breathing, dyspnea Gross lesions: mild multifocal consolidation of the lung Microscopic lesions: bronchointerstitial pneumonia, airway epithelial necrosis, type 2 pneumocyte hypertrophy and hyperplasia	[1]

Abbreviations: Ref., references.

## Data Availability

Not applicable.

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
