# Peer review of "The Effects of Swine Coronaviruses on ER Stress, Autophagy, Apoptosis, and Alterations in Cell Morphology"

_pathogens, 2022, doi:10.3390/pathogens11080940_

Round 1

Reviewer 1 Report

This is a very good mini review to introduce various swine coronavirus’ effects on host cells, particularly on ER stress, autophagy, apoptosis and changes in cell morphology, and writing is very well. Several points should be considered for modification or rewriting by authors

In section “Swine coronaviruses”, authors should provide more descriptions regarding the genomic structure of swine CoV and may add a schematic diagram of genomic structure of swine CoV.

I would like to suggest authors conduct a diagram showing the signaling pathways of Autophagy or Apoptosis induced by swine CoV infection and modulatory mechanisms, respectively, which would be helpful for our readers greatly.

Generally, describe genes should lowercase and italic, in line 42, I would suggest authors use “the s, e, m, and n genes” instead of “the S, E, M, and N genes”.

In Figure 1 legend, in line 53, “Porcine epidemic diarrhea virus (PEDV) infection, ileum, weaned pig” is an unclear statement, please clarify. I would suggest authors use arrows indicate the pathology of villous atrophy and villus fusion. And authors should denote where are the figures from in the legend? And should label where are the Peyer’s patches in b) to aid our readers to understand. What does the black label represent in b)? in addition, some other figs have the same issue. Author should carefully re-edit these figures.

Reviewer 2 Report

In this paper, authors well reviewed cellular damages and alterations caused by swine coronaviruses whose epidemic induced serious enteric, respiratory and nervous diseases, resulting in huge economic damage in pig industry. It is beneficial to understanding of pathogenesis of swine coronaviruses infection and related diseases. The paper was well organized and well written. However, some studies especially published ten years ago had been well reviewed in previous reviews. Authors do not repeat the contents but just brief introduce them by citing the reviews. There are some mistakes and language defects, need to be checked thoroughly. In addition, the number of references is too big. Authors should cut it to two third.   

Reviewer 3 Report

In the manuscript titled “The Effects of Swine Coronaviruses on ER Stress, Autophagy, Apoptosis, and Alterations in Cell Morphology”, the authors provide a comprehensive review of literature in the field of Swine Coronaviruses. The manuscript sheds light on ER stress, autophagy, apoptosis and alterations in cell morphology caused by Swine coronaviruses. The author have highlighted all the recent research articles related the topic. Here are few suggestions for the authors to improve the manuscript.

1)    Throughout the manuscript there are few sentences that are extremely short (eg.line 76) or too long. There are some typos and punctuation errors as well and there are some places with inconsistent tense use. Hence thorough editing of the manuscript would be great.

2)    Line 33-36 The authors should also cite the original paper or a review article that talks about the discovery of these viruses.

3)    There are multiple places where the authors have referred to the proteins of the swine coronavirus like N protein, nsp6. The authors should include a brief section and/or a figure about genome organization of these viruses.

4)    Line 132 the abbreviation for circular RNA is circRNA. 

5)    DEEGS should be replaced with DEGs

6)    Figure 3, though informative does not add much to overall aim of the review.

7)    Line 162 The abbreviation ERGIC used should be expanded atleast once in the manuscript.

8)    In the ER stress section the authors should also include a recent report 

“Induction and modulation of the unfolded protein response during porcine deltacoronavirus infection” Vet Microbiol. 2022

9)    In the autophagy section, lines 224-226 discussing about TGEV, the authors can consider including another report 

Doxycycline Induces Mitophagy and Suppresses Production of Interferon-β in IPEC-J2 Cells” Front. Cell. Infect. Microbiol., 01 February 2017

10) Line 310, the sentence starting “For example” needs to be reframed.

11) Three sections mentioned in the manuscript, ER stress, autophagy and apoptosis are quite complex pathways and involve action of different proteins. The readers can find it hard to keep track of these and I would suggest the authors should consider including a schematic showing the signaling pathway and the various molecules targeted by the swine coronaviruses.

12) It will also be great to have some concluding remarks or a summary for each section towards the end of each section.
